# Impact of involving people with dementia and their care partners in research: a qualitative study

Jahanara Miah [1,2] Suzanne Parsons,[2] Karina Lovell,[3] Bella Starling,[2] Iracema Leroi,[4,1] Piers Dawes[5,6]

► Prepublication history and additional materials for this paper is available online. To view these files, please visit the journal online (http://dx.doi.org/10.1136/bmjopen-2020-039321).

For numbered affiliations see end of article.

**Correspondence to**
Dr Piers Dawes;
piers.dawes@mq.edu.au

## ABSTRACT

**Objectives** We aimed to evaluate the impact of patient and public involvement (PPI) at each stage of the research cycle in a dementia research programme.

**Design** We used monitoring forms to record the impact of the research programme's PPI at different stages of research and qualitative interviews with all participants to evaluate the impact of PPI.

**Setting** We evaluated Research User Groups (RUGs—older people with dementia and care partners) which were established to provide PPI support for the research programme in multiple European sites.

**Participants** We purposively sampled RUG members (n=34) and researchers (n=13) who had participated in PPI activities. Inclusion criteria for the study were: (a) RUG members who had participated in the research awareness training and in PPI activities and had the capacity to consent; (b) researchers who involved RUGs in their work.

**Results** *Impact on the research:* changes to the study conduct were made as a result of the feedback from RUGs. These included prioritisation of clinical recommendations, the wording of study information and recruitment materials, the content and layout of the user interface for a computerised memory test, interpretation of intervention results and advice on dissemination avenues. *Impact on RUG members:* they reported that involvement had given them a sense of purpose and satisfaction. Their perception of health research changed from being an exclusive activity to one, which lay people, could have meaningful involvement. *Impact on researchers:* PPI was a new way of working and interacting with PPI members had given them insight into the impact of their work on people living with dementia.

**Conclusions** PPI can have a substantial impact on dementia research and the people involved in the research. To justify the time and expense of PPI, the advantageous practical impacts of PPI should be systematically recorded and consistently reported.

## INTRODUCTION

Recognition that considering the views of end-users in health research improves the relevance and quality of health research has raised the profile of patient and public involvement (PPI).[1–13] PPI reflects the growing democratisation of health research and the importance of research impact. PPI ensures that research questions, conduct and outcomes are relevant and meaningful to patients and the public.[11 14–18]

In countries such as the UK, PPI is required by many funders,[19] as well as an ethical obligation.[20] Researchers are encouraged to involve patients and the public at all stages of the research cycle.[21] However, substantial challenges with the conception and implementation of PPI are still evident.[22–25] Although many health research studies acknowledge the beneficial impacts of PPI,[23] various reviews highlight that the reporting of PPI is often of poor quality and evidence of impact is weak.[23–28] Thus, there is a clear need to demonstrate the impact of PPI on the research process more rigorously. Detailing the contribution made by PPI enables learning how to do PPI effectively and how to identify key indicators of impact.[29 30] In the context of the international multi-site dementia research programme, SENSE-Cog (www.sense-cog.eu), our aims were to: (a) identify the impact of PPI at each stage of the research cycle[31]; (b) provide a model for recording the impacts of PPI; and (c) explore the experiences and perceptions of people with dementia, their care partners and dementia researchers of PPI in the research programme.

### Strengths and limitations of this study

► Patient and public involvement (PPI) monitoring forms allowed systematic monitoring and evidencing demonstrable impact.
► Research User Groups (RUGs) had limited control over what level or type of PPI activities they were tasked with.
► We interviewed researchers involved in a project that contained a strong element of PPI.
► Interviews conducted by PPI coordinators may have biased RUG participant responses.
► The RUG could have included people from more diverse backgrounds.

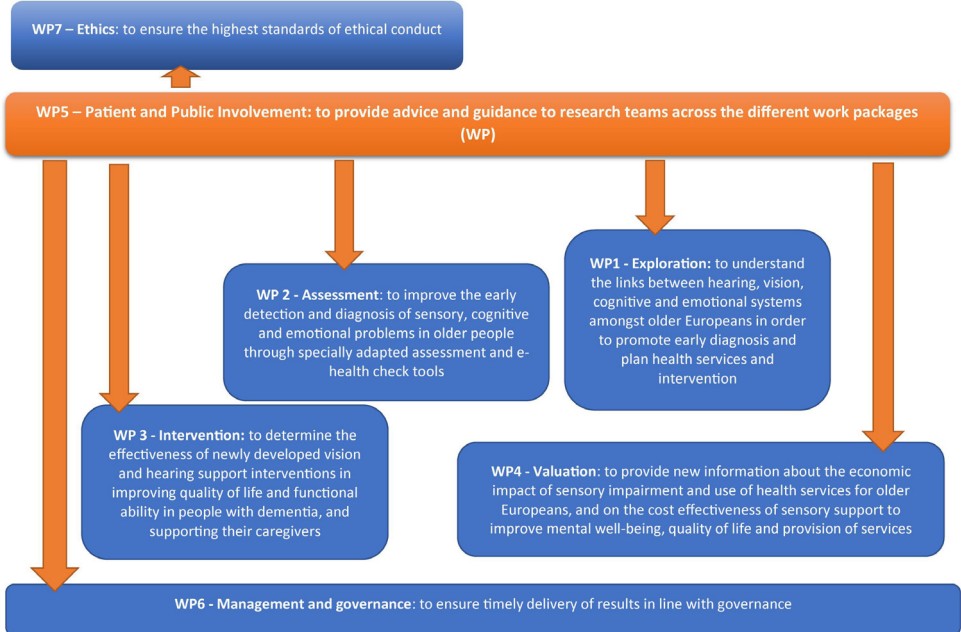

**Figure 1** SENSE-Cog work package aims. Overview of SENSE-Cog research programme investigating the combined impact of ageing-related hearing and vision impairment on mental well-being with a dedicated work package on patient public involvement embedded as a cross-cutting theme.

## Context of the SENSE-Cog PPI work

SENSE-Cog[32] is a five-year (2016–2020) European research programme investigating the combined impact of ageing-related hearing and vision impairment on mental well-being. It involves the exploration of longitudinal datasets to understand the interplay of hearing, vision and cognitive impairment,[33–35] the validation of assessments adapted for people with dementia and hearing and/or vision impairment,[36] and the evaluation of a home-based hearing and vision support intervention in improving quality of life for people with dementia.[37–39] As part of this programme, a dedicated work package on PPI[40] was embedded as a cross cutting theme (figure 1) to involve older people with dementia and care partners. This involved establishing Research User Groups (RUGs) in Manchester (UK), Nice (France), Nicosia (Cyprus) and Athens (Greece) for PPI at each stage of the SENSE-Cog programme. Each RUG consisted of seven to nine RUG members, who were people with dementia and their care partners. RUG members were provided with research awareness training (RAT; outlined elsewhere[39]) to equip them with the skills and knowledge to guide their involvement in the research and to enable them to contribute to the research programme. Each RUG site was managed and supported by a local PPI coordinator in each of the four sites.[40] We reported all features of the PPI process according to the GRIPP2 short form recommendations[30] (see online supplemental file 1) in this paper, and the full protocol involving the RUGs is published elsewhere.[40]

The published protocol[40] outlining the PPI activities pertaining to the entire SENSE-Cog programme, was established by the RUGs at the outset of the research programme. Additional PPI activities were added at different stages on an 'as needed' basis by local RUGs and to include local contextualised content for the PPI activities. These activities were coordinated both centrally by the lead PPI coordinator based in Manchester, as well as locally by the site-specific coordinators. For the basic programme-wide PPI activities, the central PPI coordination team, RUG and cross-site researchers worked together to agree on the key objectives, formulate the PPI questions, and develop and revise the materials for the PPI activities. The central PPI coordinator supported the local RUGs and their PPI coordinators to implement and shape the activities in a manner that was appropriate and relevant to their local needs.

## METHODS
### Patient and public involvement

A member of the Manchester RUG provided PPI on this paper to ensure it was informed by the perspectives of a person with experience of involvement in health research. PPI input was obtained by a face-to-face consultation. The RUG member first reviewed the structure of the paper, including a rationale, the introduction, background and methods sections. The RUG member suggested the use of diagrams to illustrate the main findings and contributed ideas on how to categorise the findings. We acted on this by using a table to illustrate the key PPI activities in this paper. Second, the RUG member reviewed and commented on the draft paper. The RUG member supported the findings and approved the final paper. PPI in this paper provided the perspective of a person with experience of involvement in health research. The RUG member requested not to be named on this paper.

**Figure 2** SENSE-Cog patientand public involvement (PPI) model of working: monitoring and evaluation. PPI model of working in monitoring and evaluation.

## Monitoring and evaluating the impact of the SENSE-Cog PPI programme

We used two methods to monitor and evaluate the PPI work package within SENSE-Cog programme, (a) PPI monitoring forms (see online supplemental file 2), throughout the research programme; and (b) qualitative interviews conducted towards the end of the programme.

### Monitoring process

For all PPI activities, local PPI coordinators kept monitoring logs of the feedback from RUG members at each site (figure 2). Types of feedback sent back to the research

teams included: the wording of an adapted rating scale evaluating support care needs; the wording of participant information sheets; content and presentation of the intervention manual; aspects of the intervention itself (a multi-module intervention involving hearing and vision rehabilitation supported by a therapist); adaptation of a hearing aid guide for person with dementia (PwD) for the intervention manual; confirmed the researchers' analysis and refined the coding framework (table 1). The completed PPI monitoring logs from the sites were sent to the central PPI coordinator in Manchester for recording

**Table 1** Summary of key PPI activities and outcomes linking to the stage of the research cycle

| PPI activities for RUGs | Outcomes due to PPI input | PPI input not acted on |
|---|---|---|
| **Identifying and prioritising the research agenda** | | |
| ► International clinical practice recommendations: recognition and management of hearing and vision impairment in dementia | ► Support for clinical practice recommendations confirmed the relevance of the work. | |
| To contribute to development, relevance and prioritisation the guidelines | ► Prioritised recommendations used in the final document for example, 'Train GP's to query if older patients have had their eyes/ears tested', 'The benefits of wearing hearing aids should be highlighted to both person with dementia (PwD) and care-partners, 'reduce burden due to communication difficulties'.<br>► Final recommendations endorsed by RUG members. | |
| **Design** | | |
| ► E-checker toolkit | ► Instructions made clearer. | ► The toolkit cannot provide a clinical diagnosis as suggested by PPI because it is an unvalidated test used for research purposes only. |
| To contribute to developing an online screening toolkit | ► Text size increased and incorporated throughout the interface for ease of access.<br>► Instructions on how to proceed test added.<br>► The layout changed to make it more spontaneous, less formal and more inviting as suggested.<br>► Feedback after each test rather than all at the end.<br>► Result feedback format changed.<br>► Female voice included.<br>► Informant questionnaire changed from 'informant' to 'a person who knows you well'.<br>► Grey boxes changed to a brighter colour.<br>► Cheering for correct answers removed. | |
| ► E-checker toolkit | ► Font and layout changed to make it easier to see and use. | |
| User test E-checker. Provide feedback on appearance, usability, the feasibility of the E-checker | ► Instructions simplified. | |
| ► Sensory support intervention<br>To comment on intervention design | ► Inclusion criteria changed to include people with more advanced dementia.<br>► The number of visits reduced to minimise the burden on PwD and care partners.<br>► Recruitment team briefed to ensure they are clear to participants about the possible benefits of participating.<br>► Recruitment team briefed to ensure that people randomised to the care as usual group know how to access support for their impairment/s. | |
| ► Paper participant information sheet (PIS) | ► Language simplified and spelling errors corrected on the paper version. | ► Change to shorten the length of the paper participant information sheet could not be made due to ethics requirements. |
| ► Audio-visual PIS | ► Supplementary text changed alongside audio-voice on audio-visual PIS. | ► The voice-over on audio-visual participant information sheet was not changed to female voice due to limited resources. |

Continued

**Table 1** Continued

| PPI activities for RUGs | Outcomes due to PPI input | PPI input not acted on |
|---|---|---|
| To provide comments on paper and audio-visual PIS | | |
| ► Public recruitment poster<br><br>To provide feedback on the presentation, readability, text, images, wording and layout of the poster | ► 'Dementia research' on title added to encourage/draw attention from target population.<br><br>► Clarified random allocation, specified as 'group 1—care as usual and group 2—intervention. | |
| ► Communication guidance | ► Communication manual supplemented with tips leaflet from action on hearing loss. | ► The communication manual was not made into a storyboard format due to limited resources. |
| To advise on contents of communication manual, and materials for use with participants and their study partners | ► Training delivered to sensory support therapists included advice about adjusting the communication manual to fit with the current abilities/stage of the PwD, to maximise their cooperation and inclusion. | |
| ► Hearing aid leaflet<br><br>Contribute to developing a dementia-friendly hearing aid user guide | ► Changed the leaflet size to A5.<br>► More white space created to make it easier to read.<br>► Font size increased.<br>► Instruction wording changed as suggested throughout the leaflet.<br>► The photo on a front-page removed and changed to hearing aid cartoon image. | |
| ► Glasses care leaflet<br><br>Contribute to developing dementia-friendly glasses care guidance leaflets | ► Changed the size, layout, font and wording of the leaflet.<br>► Cross and ticks used to indicate 'do's and don'ts'.<br><br>► The sensory support therapists (SSTs) to clarify the cloth, wipes and use of glasses straps with the participants during their visits. | |
| ► Model validation<br><br>Contribute to developing the economic model validation | ► Economic model shaped by RUG members experiences<br>► RUG members confirmed research reports concerning health economic impacts of dementia. | |
| Data gathering | | |
| ► Sensory intervention diaries<br><br>To provide feedback on participant and study partners' diary | ► Diaries were provided at each visit rather than together as a booklet at the end. | |
| ► Interview questions<br>► To advise on qualitative interview schedules on questions for PwD with hearing aids | ► Language simplified to ask about people's experiences of using hearing aids.<br>► Confirmed the relevance of the questions. | |
| ► Clinical questionnaires<br><br>To provide feedback on clinical questionnaires | ► Spelling errors corrected. | |
| ► RUD-lite questionnaire<br><br>To advise on the meaning of RUD-lite questionnaire | ► Supplementary notes on RUD-lite questionnaire added to the researcher's manual. | ► Questions could not be directly changed on the RUD-lite questionnaire as it is a validated tool for the trial. |
| Analysing and interpreting | | |

| **Table 1** Continued | | |
|---|---|---|
| PPI activities for RUGs | Outcomes due to PPI input | PPI input not acted on |
| ► Main trial qualitative data interpretation<br><br>Assist with data analysis on selected transcripts to identify codes | ► The codes identified by the RUGs confirmed the researchers' analysis and helped to refine the 'coding tree', as well as identify additional themes that were not identified by the researchers. | |
| ► Modelling the impact of sensory interventions on cognitive health | ► Discussion of RUG experiences confirmed the research team's interpretation of the results. | |
| Dissemination | | |
| ► Dissemination strategy<br><br>Advise on the dissemination plan | ► Public/social events incorporated into the dissemination plan.<br>► Local and national stakeholder groups included in the dissemination list for the newsletter and network list.<br>► YouTube videos produced online about current research programme and findings. | |
| ► The impact of sensory interventions on cognitive health<br><br>Advise on dissemination routes for the findings | ► Dissemination of findings to a wider public audience via national newspapers, including *Daily Mail*, *The Times*. | |
| ► Nicosia public engagement event<br><br>Provide feedback on content and structure of engagement event | ► Changes were made to the presentation agenda to include care partners' perspectives, information and support for care partners.<br>► 2 RUG members gave presentations about their involvement in research. | |
| ► Nice public engagement event<br><br>Provide feedback on content and structure of engagement event | ► Wording, layout and language simplified on the presentation slides.<br>► Workshop included an audience discussion on the concept of PPI. | |
| ► Newsletter | ► Newsletter named 'SENSE-Cog news'. | Newsletter published quarterly and not every 2 months as suggested due to limited resources. |
| Advise on name of the newsletter, content and format of the project newsletter | ► Suggestions concerning the content were incorporated to include updates on the studies, ongoing activities within SENSE-Cog with a section on each study site, biographic details of research teams and forthcoming events.<br>► A section on RUGs perspectives included. | |
| ► INVOLVE conference 2017<br><br>Provide feedback on a planned conference presentation | ► Changes were made to the order of the slides to make it easier to follow.<br>► Additional handout materials included for the presentation to provide more information about the research. | |
| ► PPI protocol paper<br><br>Contribute to the plain English summary section | ► Changes were made to the text to simplify the wording and the approved the plain English summary. | |
| ► PPI scoping review<br><br>Provide feedback on the main findings of the review | ► Discussions and recommendations on findings incorporated in the findings section of the paper. | |
| ► AAIC conference poster presentation<br><br>Advise on the contents and presentation of conference paper | ► Layout changed on the poster to include more white space to make it visibly appealing, wording simplified and content re-arranged to make the materials easy to read.<br>► RUGS chose photos of PPI activity to be included on the poster. | |
| ► Age and accessibility workshop<br><br>Provide feedback on a planned workshop presentation | ► More details about the support provisions for RUGs were included in the presentation slides.<br>► Wording simplified and texts size increased.<br>► Video of a RUG member talking about her experience of involvement included in the presentation. | |

 Miah J, *et al. BMJ Open* 2020;**10**:e039321. doi:10.1136/bmjopen-2020-039321

**Table 1** Continued

| PPI activities for RUGs | Outcomes due to PPI input | PPI input not acted on |
|---|---|---|
| ► NIHR PPI and inclusion workshop | ► Wording simplified. | |
| Provide feedback on a planned workshop presentation | ► Video of a RUG member talking about her experience of involvement included in the presentation. | |
| ► SENSE-Cog website publicity materials | ► Chose photographs for use in the study website and other public-facing materials. | |
| Advise on content and images for websites and publicity materials | ► Wording on the website changed to simplify language. The term 'elderly' changed to 'older European citizens' and 'seniors'. | |

NIHR, National Institute for Health Research; PPI, patient and public involvement; PwD, person with dementia; RUD-lite, Resource Utilization in Dementia-Lite version questionnaire; RUG, Research User Group.

purposes, and were then forwarded to the relevant work package research team for comments who recorded the actions taken to address the feedback on the monitoring form. All RUGs were given updates on the progress of the overall SENSE-Cog programme every 18 months to inform them of new developments and outputs and for the RUGs to witness how their feedback had shaped different aspects of the work. The PPI monitoring forms were examined by the central PPI coordinator and impacts and justifications for no action following RUG recommendations extracted and itemised in a table format (table 1).

### Qualitative interviews
#### Participants for the qualitative substudy
Participants for the impact evaluation were purposively sampled from two of the PPI working groups:

(a) RUG members (n=34) who had participated in the RAT and in PPI activities and had the capacity to consent. Establishing whether potential participants with cognitive impairment can give informed consent to was assessed by staff trained, further details on the assessment of capacity are available elsewhere[40]; (b) researchers who involved RUGs in their work.

We aimed for a sample size of six RUG participants in each RUG site and six researchers across all sites to ensure representation and was determined by data saturation, the point when no new information was obtained. We examined the coding framework data, the characteristics of the participant quotes and the findings, and

decided to end the at the point when no new information emerged but included three more interviews to check that data saturation has been reached.[41 42] Thirty-four RUG members were invited for interviews. Twenty-three RUG members responded and participated in the interviews. Eighteen researchers were approached for interviews, thirteen researchers responded and participated in the interviews. A total of 36 interviews were conducted (table 2).

#### Procedure for qualitative substudy
We wrote two interview schedules, one for RUG participants (see online supplemental file 3) and one for researchers (see online supplemental file 4). The interview schedule for RUG participants was designed to understand their experience as a RUG member, participating in PPI activities and the perceived impact of their input on the research process. The interview schedule for researcher participants was developed to explore the researcher's view of PPI, their experience of working with RUGs and their perceived impact of PPI on the research.

Interviews with RUG participants, lasting about an hour each, were conducted between October 2018 and October 2019 by local PPI coordinators in each site. The interviews were conducted separately or in pairs as preferred by the PwD and care partners. Interviews with researcher participants were conducted between January 2019 and September 2019 by the central PPI coordinator (JM) in English, either by phone of face-to-face.

**Table 2** Characteristics of participants

| | RUG members | | | | Researchers | | | |
|---|---|---|---|---|---|---|---|---|
| | Person with dementia | | Care partner | | Researcher | | Researcher project manager | |
| RUG site | Male | Female | Male | Female | Male | Female | Male | Female |
| Manchester | 3 | 0 | 0 | 4 | 0 | 5 | 1 | 0 |
| Nicosia | 2 | 1 | 1 | 1 | 0 | 2 | 0 | 2 |
| Nice | 1 | 1 | 0 | 3 | 0 | 1 | 0 | 0 |
| Athens | 2 | 1 | 0 | 3 | 0 | 1 | 1 | 0 |
| Total | 8 | 3 | 1 | 11 | 0 | 9 | 2 | 2 |

RUG, Research User Group.

Interviews with English speaking participants were audio-recorded and then transcribed verbatim. Interviews in Greek and French were transcribed verbatim in the local language and translated by the local PPI coordinators. Personal information was removed from transcripts.

### Qualitative data analysis

Researcher and RUG data were pooled across sites and analysed. We undertook thematic analysis[43] to draw out the main characteristics of RUG members and researchers experiences. JM undertook the primary analysis, reading transcripts numerous times to develop open codes. Various levels of open coding took place from line-by-line coding. JM and SP independently examined the data to identify themes and reach consensus on interpretation. Themes were then assigned codes according to the Framework method[44] using NVivo software V.11 (QSR International, Doncaster, Australia). This method allows the data to be structured into a chart with columns and rows aiding the identification and grouping of text into relevant parts.

The coding framework for the RUG participants was emailed to local site PPI coordinators who were asked to provide input on a draft of the coding framework. Any additional themes identified by coordinators were added to the list of themes. Datasets were then reanalysed according to the final coding framework. We stopped data collection when no new information was being generated from analysis.[41]

According to the Mental Capacity Act (2005),[45] people should be assumed to have capacity unless otherwise demonstrated. All researchers involved in this research completed training in assessing capacity in research. Participants were provided with easy access user-friendly versions of the participant information sheet and consent form and enough time to think about whether to participate in the study. Consent was assessed on an on-going basis. Participants provided informed written consent prior to the interview. Further information on consenting procedure is available elsewhere.[40]

## RESULTS

### Impact of PPI on research

PPI resulted in specific changes at all stages of the research from identifying and prioritising the research agenda to dissemination (table 1). In some cases, changes recommended by PPI were not implemented. Reasons that changes could not be implemented included scientific or regulatory requirements (eg, not being able to make changes to validated psychometric tests, not being able to supply clinical diagnoses on the basis of tests given for a research study, not being able to shorten the participant information sheet due to ethical governance requirements) or limited financial or personnel resources (eg, being able to produce a study newsletter only four times a year rather than more frequently).

### RUG members' perspectives

Four themes were identified: facilitators to contributing, meaningful PPI, personal gains and perceived impact of involvement on research. In the following quotes, the site is reported first (ie, 'Manchester', 'Nice', 'Nicosia' or 'Athens'). 'RUG (number)' refers to the individual RUG member being quoted. 'PwD' and 'care partner' refers to a person with dementia or a care partner for someone with dementia, respectively.

### Facilitators to contributing in PPI

Participants discussed the benefits of group work. Group work was viewed as peer support for participants, which was seen as a valuable way to help understand each other's circumstances and needs to support each other. Participants reported that they had gained new knowledge and discussed the benefits of receiving the RAT for the role, highlighting that the structuring of the training alongside the involvement tasks as factors that helped them to contribute to research. RUG participants also reported that the support provided by the local PPI coordinators made them feel that they were making a valuable contribution. A participant commented on the importance of the 'human relationship' (*Athens, RUG4, care partner*) with the coordinator.

### Meaningful PPI

Interaction with researchers and having researchers listen to RUG member's opinions to was reported as an important aspect. Receiving feedback was also highlighted as an important factor in letting RUG members know that their contribution was having a tangible impact:

> I feel that our communication was quite satisfactory and apart from that we had feedback. We said something and they 'listened' to it and we move forward, they tell you a little more, something like that …. Yes, for sure. Otherwise, I wouldn't be here today. It's a good feeling, to be heard. You gain this feeling that you are not alone and that there are people that care for you, and that what you say can go further (*Athens, RUG5, care partner*)

### Personal benefits of PPI

Face-to-face group meetings provided RUG members the opportunity to share issues with others in similar position. RUG members reported that the experience personally benefited them in terms of gaining new knowledge, interacting with others, giving advice and feeling that they were making a worthwhile contribution. RUG participants felt their involvement exceeded their expectations, due to continuous involvement and training in research awareness:

> Throughout the meetings we felt that what we learned is very important for the lives of individuals. I felt the role was more than was described (role of RUG member), because we participated (learned) in many task trying to find solutions that we have to resolve. (*Athens RUG6, PwD*)

## Perceived impact of involvement on research

RUG members talked about their involvement in dissemination activities, particularly in countries other than the UK. RUG members talked about how providing their perspective provided researchers with a better understanding of the impact of dementia that informed research decisions. RUG members emphasised their contribution in making written materials more accessible for participants and suggesting changes in the interventions and tools developed by the research team. RUG members reported that incremental changes informed by PPI could make a difference to practical outcomes:

> Well it's just one factor in the complicated chain of conditions that lead to the outcome. So it's not going to have any direct impact on any individual people, but every time a member of the group makes a contribution, it's adding a little bit more information to the whole package of information that's going to improve the lives of the end users in the long term. (*Manchester, RUG1, ex-care partner*)

### Researchers' perspectives

Four themes were identified: limited experience of PPI, acceptance and importance of PPI, beneficial impacts of involvement and challenges. These are presented in table 3, in the quotes, the site is reported first (ie, 'Manchester', 'Nice', 'Nicosia' or 'Athens') then the researcher number. 'PM' refers to project managers.

## DISCUSSION

PPI is proposed as an important means of enhancing the relevance and impact of research.[21 46] However, while the descriptive literature on PPI is growing, there is still little evidence for how PPI actually impacts on research and there has been no clear methodology for recording impact.[22–24 27] Here, we reported a method for capturing the impact of PPI in a multi-centre European research project, outlining the impact of PPI at each stage of the research process. We also provided qualitative perspectives of RUG members and researchers concerning their views on the impact of involving people with dementia and care partners in research.

**Table 3** Researchers' perspectives on impact of involvement

| Themes | Item |
| --- | --- |
| Limited experiences of PPI | Of the 13 researchers interviewed, 10 had no previous experience of systematically obtaining patient and public input to planning or conduct of research: |
| | 'So it was the first time we've heard about it, on this project. So it took us all a while to understand it, but I mean by going through it and by doing it we understood the importance of it'. (*Nicosia, R3*) |
| Acceptance and importance of PPI | There was general acceptance of PPI by researchers and acknowledgement of the importance of the PPI integrated into the research programme. Researchers reported that PPI provided them with the perspective of living with dementia: |
| | 'I think it's very useful and I think it gives a perspective to research that we needed to have, because we kind of tend to, you know, sort of take the feeling out of our day to day research and when you have the actual patients explain to them rather than peers, it does give us a different perspective' (*Manchester, PM9*) |
| Beneficial impacts of involvement | Researchers talked about the beneficial impacts of PPI, including refining content and wording of public-facing information and identifying issues not considered by the researchers. One researcher commented that PPI provided insight into what patients and the public think of their work. Researchers valued PPI with respect to understanding the impacts of dementia on daily life and how these impacts should be captured in health economic measurement, ensuring the appropriateness and relevance of health economic measures: |
| | 'We were trying to confirm that using dependence as some measure of progression in our economic model would be appropriate and having spoken to the RUG members, they, kind of, confirmed that. And then there were some, kind of, hallmark things that stood out that they all, kind of, agreed on in their discussions … It has helped in terms of model of validation'. (*Man, R12*) |
| | Some researchers also talked about the emotional impact of listening to RUG members' experiences, which were distressing at times, particularly for researchers not used to patient contact. Researchers reported that their interaction with RUG members led to researchers changing future practice, such as taking patient perspectives into account in planning and prioritising research: |
| | 'I've written numerous grant proposals since SENSE-Cog began, and in some of them I've included … wherever it makes sense, I've included a research utilisation for a patient and public involvement group'. (*Nicosia, PM8*) |
| Challenges | Researchers talked about challenges, such as PPI suggestions not being able to be implemented because of scientific reasons or resource limitations. Another researcher highlighted the importance of identifying points of research design where PPI would have a critical impact and reported a missed opportunity concerning selection of outcome measures for an intervention study due to time constraints in development of the study protocol. |

PPI, patient and public involvement; RUG, Research User Group.

A number of key changes were made to different aspects of the research programme at various points following PPI advice. PPI contributed substantive input to intervention formation, design, recruitment, interpretation of results and dissemination of the research. However, some PPI feedback could not be acted on for the reasons outside the research team's control due to resource restrictions, lack of time or compromising validated psychometric measures. These restrictions on implementing PPI advice are similar to restrictions reported elsewhere.[15 26] Without ongoing advice from the RUGs to the research teams, researchers may not have made appropriate changes on many aspects of the research design to take account of the experience of living with dementia. In addition to concrete changes in the research following RUG input, RUG members reported various personal benefits in contributing to research, including feeling that they were making a useful contribution to research. Researchers reported that the RUG members input gave them an important personal perspective on the relevance and impact of their research for people living with dementia.

### Limitations

RUG interview data were collected by the coordinators who were themselves SENSE-Cog project researchers, rather than by an independent researcher. Data collection by SENSE-Cog project researchers may have resulted in responses being overly positive.[43 44]

Although research involvement opportunities for RUGs were advertised via contact with local charities, social organisations and support groups, we received few queries about involvement and no queries from anyone from minority ethnic communities. The RUG groups may not have reflected the diversity within the general community, and this may have limited the perspectives that RUGs were able to offer and of the impact of PPI in this project. Using local community leaders or organisations to access more diverse communities could have increased recruitment from minority groups.[47–49] Future research could explore views of older adults from minority ethnic background and diverse communities in PPI.

For our qualitative work, we only sought the opinions of researchers who were involved in a project that contained a strong element of PPI. The researchers in this project may have been more favourably disposed to PPI than European researchers in general. Another limitation is that RUGs were not involved in the initial conception of the study at the grant application stage. Ideally, PPI should occur from the point of conception of a research programme and around setting research agendas. Due to time and resource constraints, and possible extra burden we did not involve RUG members in data collection although they were involved in the development of the interview questions.

It is appropriate to highlight the issue of the power imbalance[13] between researchers and RUGs. The RUGs had limited control over what level or type of PPI activities they were tasked with in the research programme. In addition, RUG members had limited influence on what feedback were acted on, as they were dependent on researchers listening to their input and acting on their feedback. We provided RUG members with notes of the PPI activities to acknowledge their contribution and provided written responses from the researchers on what actions were taken because of the RUGs contributions.

PPI in research has potential benefits, but it does require substantial resources that could otherwise be spent elsewhere. Although this paper highlighted the impact of PPI on the research and documented positive experiences of the RUG members and researchers, we cannot be sure of the impact of these changes on research outcomes (eg, if fewer participants would have been recruited if recruitment materials had not been altered following PPI input). An understanding of the benefit and cost utility of PPI would require a controlled evaluation (eg, recruitment with vs without PPI input on recruitment materials). There is some evidence from controlled evaluations that PPI does have tangible impact: a systematic review on the impact of PPI on enrolment and retention in clinical trials reported that PPI considerably improved the likelihoods of a patient joining a trial compared with recruitment strategies that did not include PPI.[18] However, because recruitment strategies that had used PPI also included non-PPI components, it was difficult to establish the contribution of PPI versus other factors.

The overall strength of this study is the demonstrable impacts of PPI on research and the benefits reported for both researchers and RUGs interactions highlight gains in experiential knowledge and learning.

### CONCLUSION

This study provides a methodology for recording the impact of PPI in multi-centre health research using systematic ongoing monitoring and feedback throughout the research programme, followed by detailed qualitative exploration of PPI contributors' and researchers' views. To justify the time and expense of PPI, the impacts and benefits of PPI should be systematically recorded and reported for all health research that makes use of PPI.

Further information about SENSE-Cog is available in online (www.sense-cog.eu).

**Author affiliations**
[1]Division of Neuroscience and Experimental Psychology, University of Manchester, Manchester, UK
[2]Public Programmes Team, Manchester University NHS Foundation Trust, Manchester, UK
[3]The School of Nursing, Midwifery and Social Work, The University of Manchester, Manchester, UK
[4]School of Medicine, Global Brain Health Institute, Dublin, Ireland
[5]Manchester Centre for Audiology and Deafness (ManCAD), University of Manchester, Manchester, UK
[6]Department of Linguistics, Australian Hearing Hub, Macquarie University, Sydney, New South Wales, Australia

**Acknowledgements** The authors would like to acknowledge the people working with us as part of Research User Groups and SENSE-Cog colleagues coordinating the Research User Groups in the UK, France, Cyprus and Greece.

**Contributors** IL, PD, KL, BS contributed to the conception of the study. JM, PD and SP contributed to the design of the study. JM wrote the first draft and was involved in the editing of the manuscript, which was critically reviewed by IL, KL, PD and SP and approved the final manuscript.

**Funding** This protocol paper is part of Work Package 5 of the SENSE-Cog project, which has received funding from the European Union's Horizon 2020 research and innovation programme under grant agreement No. 668648. Piers Dawes, Suzanne Parsons and Bella Starling are supported by the National Institute for Health Research Manchester Biomedical Research Centre.

**Disclaimer** The funding body has no part in study design, data collection, data analysis, data interpretation or manuscript writing.

**Competing interests** None declared.

**Patient consent for publication** Not required.

**Ethics approval** Ethical Approval was approved by the University of Manchester's Research Ethics Committee UREC: 2017-0627-2142. Additional ethical approvals were sought and obtained for each study site; in Nicosia approved by the Cyprus National Bioethics Committee, in Nice by 'Comité de Protection des personnes Sud Est I', and in Athens by Ethical Committee of Health Sciences and Scientific Committee of the Eginition Hospital of the National and Kapodistrian University of Athens.

**Provenance and peer review** Not commissioned; externally peer reviewed.

**Data availability statement** Interview data from this study are personally identifiable for RUG members and researchers, and are therefore not available for data sharing.

**ORCID iD**
Jahanara Miah http://orcid.org/0000-0002-2122-7007

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
