## [Reviewer comments · BMJ Open]

ARTICLE DETAILS

TITLE (PROVISIONAL)	The impact of involving people with dementia and their care partners in research: A qualitative study
AUTHORS	Miah, Jahanara; Parsons, Suzanne; Lovell, Karina; Starling, Bella; Leroi, Iracema; Dawes, Piers

VERSION 1 – REVIEW

REVIEWER	Prof Brian Taylor Ulster University, Northern Ireland
REVIEW RETURNED	22-May-2020

GENERAL COMMENTS	1. Clarity of purpose and title The title of the manuscript accurately conveys the purpose of the study. The word 'impact' in the title might commonly be questioned for a qualitative study, but this paper reports on the changes made to the study protocol as a result of the co-production initiative, and hence this word in the title seems appropriate in this case. 2. Originality and Contribution to Knowledge This paper adds to our understanding of identifiable effects of involving people with dementia in qualitative research (which in this case was a multi-site study of the impact on mental well-being of ageing-related hearing and vision impairment). The paper provides details of a wide range of recommendations of the research user group, and tabulates the ensuing amendments to the main study. It would be helpful to readers seeking a rounded understand of the topic if the main paper included the references in the protocol to other studies on involving people with dementia as peer-researchers, as this is the focus of the paper. 3. Context and Literature The context and coverage of the literature shows depth across published articles on involving people with dementia as co-researchers. Key sources such as the INVOLVE organisation in the UK are cited. 4. Quality of Methods and Results The method is clearly explained and the design is appropriate for the purpose. Data gathering tools are appropriate and the data gathering appears sound. Appropriate ethical approval for the study is cited. Table 1 summarising Personal and Public Involvement activities and outcomes has headings linking findings from the study to stages of the research cycle. It would be helpful to cite the source for these definitions of research stages. In particular, this Table would be more generalisable for readers if there were a section for 'data gathering' or 'data gathering tools', which is a common sub-heading within the methods section of research papers. A Supplementary File very helpfully tabulates the Findings against the revised Guidance for Reporting Involvement of Patients and the Public in research (GRIPP2).
--

	5. Discussion and Conclusions The limitations of the study are clearly stated. It would be helpful if the limitations section acknowledged also the risks that may be inherent in involving service users in research, such as identified in the study in paper 44 in the study protocol. It would be helpful to note that the study did not include consideration of involving research users in the actual gathering of qualitative data through interviews or focus groups (see eg. Taylor et al., 2014), although there were comments on proposed interview questions. This is understandable in that the primary methodology of the main study involves secondary data. However a slight restructuring of the categories in Table 1 (as commented above) would highlight which elements of data gathering methodology were considered in this study (such as the comments on the sensory intervention diaries), and which were not. The argument is logically constructed and the discussion is balanced. The conclusions are supported by the data presented. The article includes honest reflective comments about the emotional impact of the co-production process on some of the researchers (who are presumably not professionals familiar with working with this client group). 6. Organisation of manuscript The manuscript is well structured, and contains valuable supplementary files giving details of the methods used. 7. Abstract and Keywords The abstract is clear, and contains a summary of the context, method, results and discussion. These accurately convey the work. The keywords are appropriate for digital database searching, but could usefully include also a term relating to 'co-production' (of research) or 'patient and public involvement' or 'patient and client involvement' or similar as this is central to the topic of the paper. 8. Quality of writing, referencing & visual material The quality of writing and referencing is generally of a high standard. Is the American spelling of 'program' in the abstract appropriate for this UK-based journal? There is a typo in line 100: 'RUGS' should be 'RUGs'. 9. Relevance & interest to the journal readership This paper is relevant and of interest to the readership of the journal. 10. Conclusion This paper makes a valuable contribution to furthering the involvement of people with dementia in co-production of research. Reference Taylor BJ, Killick C, O'Brien M, Begley E & Carter-Anand J (2014) Older people's conceptualisation of elder abuse and neglect. Journal of Elder Abuse and Neglect, 26(3), 223-243. https://doi: 10.1080/08946566.2013.795881
--	--

REVIEWER	Rabih Chattat University of Bologna, Italy
REVIEW RETURNED	09-Jun-2020

GENERAL COMMENTS	Dear Authors. Thanks for addressing such an important issue in research. Abstract: The abstract is well structured I would suggest to add some information about participants (RUG are people with dementia and caregivers) and on tools used for monitoring in order to allow readers to get more information about the study. Introduction: In the introduction the authors refer to the most important literature regarding the topic of PPI. The study objective (research questions are clear and well addressed.
---

	Methods: The methods are clearly described. In the section (participants for the qualitative sub-study p.6 line 145-153) the authors set that they involve 6 participant by site for a total of 34 without clarifying if they are related to People with dementia or to caregivers. Furthermore no information are given about the people with dementia participating at the study. They mention the capacity to consent but what they mean by and how they assess the capacity to consent. On the other hand, authors aim is to recruit a sample size of 6 RUG members by site but they refer also to data saturation which usually mean, as mentioned by authors that recruitment will continue until no new information are obtained. I would suggest to authors to give more info about how they proceed. Authors refer frequently to supplementary file but some more info in the text can allow reader a better understanding of the study. (see for example p6 line 159). Results: In the results regarding RUG members perspectives the point of view of people with dementia and thir caregivers are grouped together. Is there any differences between people with dementia and their caregivers. It would be interesting if differences can be put in evidence. Table 1 offer detailed information about the results of the study and how authors manage to include PPI information and the adaptation used. The results regarding researchers perspective can be shortened also in a table and may reporting if some adaptation or initiative had been undertaken toward. Discussion: On line 320 (p.20) the authors highlight the contribution of PPI to the conception and design while on p21 line 347 they report that a limitation of the study is that “RUGs were not involved in the initial conception or planning of the study. It seem that RUGs are involved during the execution phase of the study rather than in planning. I would suggest authors to be more precise about. The section regarding limitation may authors should report also the strength of the project already reported at the beginning of the paper.
--	--

VERSION 1 – AUTHOR RESPONSE

Reviewer: 1

1. Clarity of purpose and title

The title of the manuscript accurately conveys the purpose of the study. The word 'impact' in the title might commonly be questioned for a qualitative study, but this paper reports on the changes made to the study protocol as a result of the co-production initiative, and hence this word in the title seems appropriate in this case.

2. Originality and Contribution to Knowledge

This paper adds to our understanding of identifiable effects of involving people with dementia in qualitative research (which in this case was a multi-site study of the impact on mental well-being of ageing-related hearing and vision impairment). The paper provides details of a wide range of recommendations of the research user group, and tabulates the ensuing amendments to the main study. It would be helpful to readers seeking a rounded understand of the topic if the main paper included the references in the protocol to other studies on involving people with dementia as peer-researchers, as this is the focus of the paper.

Response - Thank you for the suggestion. We have now included references 2-13 and 14-18 (from the protocol paper) to provide the reader with a rounded understanding of the topic covered in the main paper.

3. Context and Literature

The context and coverage of the literature shows depth across published articles on involving people with dementia as co-researchers. Key sources such as the INVOLVE organisation in the UK are cited.

4. Quality of Methods and Results

The method is clearly explained and the design is appropriate for the purpose. Data gathering tools are appropriate and the data gathering appears sound. Appropriate ethical approval for the study is cited. Table 1 summarising Personal and Public Involvement activities and outcomes has headings linking findings from the study to stages of the research cycle. It would be helpful to cite the source for these definitions of research stages. In particular, this Table would be more generalisable for readers if there were a section for 'data gathering' or 'data gathering tools', which is a common sub-heading within the methods section of research papers. A Supplementary File very helpfully tabulates the Findings against the revised Guidance for Reporting Involvement of Patients and the Public in research (GRIPP2).

Response – Thank you for this observation and suggesting the addition of 'data gathering' section. We have cited the reference for the research cycle in Table 1 and a subheading for data gathering included in table 1, with details of PPI activities.

5. Discussion and Conclusions

The limitations of the study are clearly stated. It would be helpful if the limitations section acknowledged also the risks that may be inherent in involving service users in research, such as identified in the study in paper 44 in the study protocol.

It would be helpful to note that the study did not include consideration of involving research users in the actual gathering of qualitative data through interviews or focus groups (see eg. Taylor et al., 2014), although there were comments on proposed interview questions.

This is understandable in that the primary methodology of the main study involves secondary data. However a slight restructuring of the categories in Table 1 (as commented above) would highlight which elements of data gathering methodology were considered in this study (such as the comments on the sensory intervention diaries), and which were not. The argument is logically constructed and the discussion is balanced. The conclusions are supported by the data presented. The article includes honest reflective comments about the emotional impact of the co-production process on some of the researchers (who are presumably not professionals familiar with working with this client group).

Response – Thank you for the suggestions/references. We have acknowledged the inherent risks associated with involving service users in research in the limitations section and noted that we did not include research users in the data collection.

6. Organisation of manuscript

The manuscript is well structured, and contains valuable supplementary files giving details of the methods used.

7. Abstract and Keywords

The abstract is clear, and contains a summary of the context, method, results and discussion. These accurately convey the work. The keywords are appropriate for digital database searching, but could usefully include also a term relating to 'co-production' (of research) or 'patient and public involvement' or 'patient and client involvement' or similar as this is central to the topic of the paper.

Response-Thank you for the suggested terms, co-production is included in the keywords.

8. Quality of writing, referencing & visual material

The quality of writing and referencing is generally of a high standard. Is the American spelling of 'program' in the abstract appropriate for this UK-based journal? There is a typo in line 100: 'RUGS' should be 'RUGs'.

Response - Thank you for highlighting this, 'program' corrected to UK 'programme'

9. Relevance & interest to the journal readership
This paper is relevant and of interest to the readership of the journal.

10. Conclusion
This paper makes a valuable contribution to furthering the involvement of people with dementia in co-production of research.

Response – Thank you

Reviewer: 2

Dear Authors.
Thanks for addressing such an important issue in research.

Abstract: The abstract is well structured I would suggest to add some information about participants (RUG) are people with dementia and caregivers) and on tools used for monitoring in order to allow readers to get more information about the study.

Response – Thank you for the suggestion, we have included the information about RUGs and on tools in the abstract.

Introduction: In the introduction the authors refer to the most important literature regarding the topic of PPI. The study objective (research questions are clear and well addressed).

Methods: The methods are clearly described. In the section (participants for the qualitative sub-study p.6 line 145-153) the authors set that they involve 6 participant by site for a total of 34 without clarifying if they are related to People with dementia or to caregivers. Furthermore no information are given about the people with dementia participating at the study. They mention the capacity to consent but what they mean by and how they assess the capacity to consent. On the other hand, authors aim is to recruit a sample size of 6 RUG members by site but they refer also to data saturation which usually mean, as mentioned by authors that recruitment will continue until no new information are obtained. I would suggest to authors to give more info about how they proceed. Authors refer frequently to supplementary file but some more info in the text can allow reader a better understanding of the study. (see for example p6 line 159).

Response: We have included a reference to table 2 which provides a breakdown of PwD and caregivers, and researchers. In addition, we have included a break down of the gender in the table, which maybe be helpful for the reader. Unfortunately, we did not collect additional demographic details of our RUG members, as initially, we found it difficult to recruit for this group. We were advised by our focus group (information in protocol paper) that we do not specify dementia diagnosis on RUG recruitment flyers/advertisements as it is a sensitive issue and makes people feel uncomfortable particularly for PPI activities. The dementia conditions were self-reported, and the RUG membership criteria for inclusion was to have lived experiences of dementia.

Due to the word limit, we did not include details on how we ensured the capacity to consent or on-going consent but have include some details and referred to our protocol paper, which explains this in detail.

We have included details on how we checked that data saturation has been reached.

Thank you for the suggestion to include additional information in the main paper, we have included the interview topic guides for RUG members and researchers as figures 3 &4 in the main paper.

Results: In the results regarding RUG members perspectives the point of view of people with dementia and their caregivers are grouped together. Is there any differences between people with dementia and their caregivers. It would be interesting if differences can be put in evidence. Table 1 offer detailed information about the results of the study and how authors manage to include PPI information and the adaptation used. The results regarding researchers perspective can be shortened also in a table and may reporting if some adaptation or initiative had been undertaken toward.

Response – Thank you for the suggestions. There was no significant difference in the opinions between the people with dementia and their caregivers, therefore we grouped these. As suggested, we have now presented researchers perspectives in a table format, which has helped manage the word limit.

Discussion: On line 320 (p.20) the authors highlight the contribution of PPI to the conception and design while on p21 line 347 they report that a limitation of the study is that “RUGs were not involved in the initial conception or planning of the study. It seem that RUGs are involved during the execution phase of the study rather than in planning. I would suggest authors to be more precise about. The section regarding limitation may authors should report also the strength of the project already reported at the beginning of the paper.

Response – Thank you for the observation. We have clarified that the RUGs were not involved at the agenda/priority setting stage for the grant application. Whereas input was substantial in the intervention development stage. We have included the strengths reported in the abstract in the limitations section